# A VLP-Based Vaccine Candidate Protects Mice against Japanese Encephalitis Virus Infection

**DOI:** 10.3390/vaccines10020197

**Published:** 2022-01-26

**Authors:** Limin Yang, Aibo Xiao, Hu Wang, Xiaojuan Zhang, Yuan Zhang, Yunlong Li, Yanqiu Wei, Wenjun Liu, Chuangfu Chen

**Affiliations:** 1College of Animal Science and Technology, Shihezi University, Shihezi 832003, China; lmyang@im.ac.cn; 2CAS Key Laboratory of Pathogenic Microbiology and Immunology, Institute of Microbiology, Chinese Academy of Sciences, Beijing 100101, China; zy951110xx@163.com (Y.Z.); leeclouddargon@163.com (Y.L.); weiyanqiu0611@126.com (Y.W.); liuwj@im.ac.cn (W.L.); 3Liaoning Agricultural Development Service Center, Shenyang 110003, China; 13940547645@126.com (A.X.); happylife0327@163.com (H.W.); 4Panjin Modern Agriculture Development Center, Panjin 124010, China; pjzhangxiaojuan@126.com; 5University of Chinese Academy of Sciences, Beijing 101409, China

**Keywords:** Japanese encephalitis, VLP, envelope, yeast, cellular immune

## Abstract

Japanese encephalitis virus (JEV) is the leading cause of epidemic encephalitis in Asia, and vaccination is the most effective way to prevent JE. Although several licensed vaccines were widely used, there is still a demand for developing safer, cheaper, and more effective JE vaccines. In the current study, a virus-like particle (VLP) vaccine candidate containing the envelope structural protein of JEV expressed by the *Pichia pastoris* was assembled in vitro. It elicited a robust humoral and cellular immune response in mice model, conferring immunodeficient mice complete protection against lethal doses of JEV challenge. Furthermore, pigs immunized with VLP alone without adjuvant via intramuscular produced high neutralizing antibodies against JEV. Consequently, this study showed a new design of JEV subunit vaccine based on VLP strategy and demonstrated the potential for clinical application.

## 1. Introduction

Japanese encephalitis (JE) is a zoonotic mosquito-borne disease caused by JE virus (JEV). JEV belongs to the genus *Flavivirus* and is transmitted between pigs and humans by culicine mosquitoes [1]. Pigs serve as amplifying hosts, and humans are dead-end hosts [2]. The first confirmed JE case was reported in Japan in 1924. JE is mainly endemic in Asia and adjacent regions, and it has been gradually spreading to other territories. It is estimated that there are about 50,000–175,000 people infected with JEV, resulting in 15,000 deaths annually, and about 60% of the global population lives at risk of exposure to JEV [3,4]. JE can lead to central nervous system injury and long-term neurological, psychological, and cognitive impairment sequelae, with a mortality rate of 5–40% [5]. In addition, JEV will lead to abortion, stillbirth, congenital disabilities, and fatal neurological disease in pig herds, causing considerable losses to the pig industry every year [6]. Therefore, it is of great significance to control the prevalence of JEV.

There is no antiviral intervention to treat JE, and vaccination is the only strategy to develop long-term sustainable protection against JEV infection. There are four types of licensed JE vaccines: mouse brain or cell culture-derived inactivated vaccine, live-attenuated vaccine, and recombinant live-attenuated chimeric vaccine. Mouse brain-derived vaccine (JE-VAX) was used in many countries for decades, but due to a certain incidence of side effects (mainly including hypersensitivity reactions), production was discontinued in 2005. The live-attenuated vaccine, SA-14-14-2, developed by China, demonstrated excellent safety and effectiveness (88–96%), and more than 1300 million doses were administrated in Asia [5]. No obvious side effects were reported so far. However, this vaccine was not yet used in multiple developed countries, including the United States, due to potential safety risks. A cell culture-derived inactivated vaccine (IC51) based on SA-14-14-2 virus strain was licensed in the United States, European Union, Japan, South Korea, etc. The chimeric vaccine (ChimeriVax-JE) was generated by inserting the precursor membrane (prM) and envelope (Env) genes of SA-14-14-2 into Yellow fever (YF) 17D viral “backbone” to form a live-attenuated vaccine [7]. In addition, there are several vaccine candidates based on different strategies in preclinical research, including DNA vaccine [8,9,10], peptide and protein subunit vaccines [11,12,13], replication-defective vaccine [14], and virus vector vaccines [15].

The JEV consists of single, positive-stranded RNA and three structural proteins: capsid (C), prM, and Env. The C protein combines with RNA to form the nucleocapsid. The prM is closely associated with Env protein and act as a chaperon to promote Env maturation. Env protein functions in host cell receptor binding, viral entry, and it is the major target for humoral immunity and vaccine design. Here, we designed and produced a VLP based vaccine candidate against JEV using prM/Env protein expressed by *Pichia pastoris*. The JEV VLP demonstrated good immunogenicity in mice and pigs. Notably, it conferred complete protection against JEV challenge in the immunodeficient mice model. The data support further translational studies to advance the development of this vaccine candidate.

## 2. Materials and Methods

### 2.1. Cells, Virus, and Animals

Vero cells (ATCC CCL-81) were grown in Dulbecco’s modified Eagle’s medium (DMEM) supplemented with 10% fetal bovine sera and 100 U/mL penicillin-streptomycin (Gibco, Grand Island, NY, USA) at 37 °C in 5% CO_2_. *Aedes albopictus* C6/36 cells (ATCC CRL-1660) were cultured in RPMI 1640 medium containing 10% FBS at 28 °C CO_2_-free incubator. The yeast cell (*Pichia*
*pastoris* strain X33) and expression plasmid pPICZA were obtained from Invitrogen (Carlsbad, CA, USA).

The JEV SA 14-14-2 strain, isolated from live-attenuated JEV vaccine manufactured by Chengdu Institute of Biological Products Co. Ltd. (Sichuan, China), was conserved in IMCAS. The JEV were propagated in C6/36 cells, titrated by standard plaque-forming assay on Vero cells, and stored at −80 °C.

BALB/c mice were purchased from Beijing Vital River Laboratory Animal Technology Co., Ltd. (Beijing, China). Immune-deficient A129 mice were purchased from the Institute of Laboratory Animal Science, Chinese Academy of Medical Sciences and Peking Union Medical College. Pigs were provided from Zhangwu Zhengcheng Pig Breeding Co., Ltd. Animals were randomly allocated to groups. All animal studies were performed blinded.

### 2.2. Gene Construction

The JEV SA 14-14-2 prM/Env gene (GenBank access: AF315119.1) comprising the stem (ST) but not the transmembrane (TM) regions was synthesized by GenScript (Nanjing, China) using *Pichia pastoris* codon-optimized sequence for enhanced expression. The modified prM/Env gene was then cloned into *Pichia* expression vector pPICZA under the control of the inducible *AOX1* promoter. In this way, we obtained a *Pichia* expression plasmid that expresses prM/Env recombinant protein. The JEV prM/Env gene encodes the truncated Env protein that comprises 456 amino acids (residues Phe_1_–Met_456_), preceded by the C-terminal 33 amino acids of the prM protein (residues Ala_135_–Ser_167_) to ensure proper post-translational processing of Env. The construction contains a C-terminal 6 × His tag to facilitate purification.

### 2.3. Generation of JEV VLP Vaccine Candidate

The expression plasmid was subsequently linearized with endonuclease *Sac*I and integrated into the host genome of *Pichia*
*pastoris* strain X33 by electroporation according to the manufacturer’s instructions. Positive transformants were subsequently confirmed by PCR to assess the gene copy number integrated into the *Pichia*
*pastoris* genome. The resulting transformants were cultured to logarithmic phase in buffered complex glycerol medium (BMGY) at 30 °C and then induced using 0.5% methanol at 12 h intervals for 72 h. Induced cells were collected by centrifugation and then lysed by ultrahigh-pressure homogenization (FB-110Q, Shanghai li tu, Shanghai, China) with three cycles at 1200 bar in precooled cell lysis buffer (50 mM sodium phosphate, 1 mM PMSF, 1 mM EDTA and 5% glycerol, pH 7.4). The lysates were subjected to centrifuge for 40 min at 12,000 rpm, 4 °C, separated into supernatant (S) and pellet (P) fraction, analyzed by western blotting with mouse antisera against JEV to confirm the expression of recombinant Env. The membrane-enriched P fraction was solubilized in membrane extraction buffer (50 mM Tris-HCl, 500 mM NaCl, 6 mM GuHCl, 20 mM imidazole, pH 8.0) and clarified by centrifugation and filtration (0.22 µm). The recombinant protein was purified from the dissolved P fraction by Ni-NTA affinity chromatography (GE Healthcare, Chicago, IL, USA). Purified protein was then analyzed by SDS-PAGE and western blotting.

To assemble JEV VLP in vitro, the purified protein was first diluted with precooled assembly buffer (50 mM Tris-HCl, 50 mM NaCl and 0.01% Tween-20, pH 8.0) to a concentration of 200 µg/mL, and then dialyzed against assembly buffer for 24 h at 4 °C with twice buffer exchange. The self-assembled VLP was purified by ultracentrifugation through a 30–60% sucrose density gradient at 100,000 g for 4 h at 4 °C. To visualize the formation of particulate structure, the resulting VLP was negatively stained with 1% uranyl acetate solution and examined by JEM-1400 transmission electron microscope (JEOL Ltd., Tokyo, Japan).

### 2.4. Western Blotting

Purified recombinant Env protein was first separated by SDS-PAGE, and then electro-transferred to polyvinylidene difluoride (PVDF) membrane using a semi-dry blotting apparatus (400 mA, 40 min). After transfer, the membrane was blocked with 5% skim milk in TBS buffer containing 0.1% Tween-20 for 2 h at 37 °C, rinsed once with PBST. Afterward, the membrane was incubated with primary antibody (mouse polyclonal antibodies against JEV Env at 0.1 µg/mL, prepared in blocking buffer). After 1 h incubation, the membrane was washed five times with TBS containing 0.1% Tween-20 and then incubated with horseradish peroxidase (HRP)-conjugated goat anti-mouse IgG (0.1 µg/mL prepared in blocking buffer) as the secondary antibody for one hour, and then washed five times. Finally, the membranes were detected by ECL system (CLINX, Shanghai, China).

### 2.5. Immunization

#### 2.5.1. BALB/c Mice

Three groups of 6-week-old BALB/c female mice (*n* = 5 per group) were vaccinated subcutaneously (s.c.) with 10 μg of JEV-VLP or 10^4^ TCID_50_ of SA 14-14-2 live-attenuated vaccine, or an equivalent volume of PBS as a sham control at weeks 0, 3, and 6. Sera were collected at weeks 0, 3, 6, and 7 for ELISA and neutralizing antibodies (NAbs) test. To investigate the cellular response after vaccination, the vaccinated mice were euthanized, and spleen lymphocytes were collected for IFN-γ enzyme-linked immunospot (ELISPOT) assay.

#### 2.5.2. Pigs

Five groups of 4-week-old piglets (*n* = 3 per group) were vaccinated intramuscularly (IM) with four different doses, 2.5 μg, 5 μg, 10 μg, and 20 μg of JEV VLP or PBS control at weeks 0, 3, and 6. Sera were collected at weeks 0, 3, 6, 9 and 12 to test antibody response by ELISA and NAbs assay.

### 2.6. ELISA

JEV-Env-specific ELISA was used to determine endpoint binding antibody titers of immune sera. Endpoint titers were defined as the reciprocal sera dilution that yielded an OD450 > 2-fold over background values. Briefly, 96-well plates were coated with 10 μg/mL recombinant prM/Env protein in carbonate-bicarbonate buffer, pH 9.6, at 4 °C overnight. The plates were then blocked with 5% skim milk in PBS (pH 7.4) at 37 °C for 1 h. Sera samples were added to the top row (1:40), and 2-fold serial dilutions were tested in the remaining rows. The plates were incubated at 37 °C for 1 h, followed by five washes with PBST. Subsequently, the plates were incubated with an HRP-conjugated antimouse antibody working solution at 37 °C for 30 min and washed with PBST five times. The assay was developed using 3,3′,5′,5-Tetramethylbenzidine HRP substrate (TMB) with 100 µL each well stopped by adding 50 µL of 2 M H_2_SO_4_ for 10 min. Plates were measured at 450 nm by a microplate reader using Softmax Pro 6.0 software (Molecular Devices, San Jose, CA, USA).

### 2.7. Neutralization Assay

JEV-specific NAbs were measured with a standard 50% plaque reduction neutralization test (plaque-reduction neutralization test, PRNT) as described previously with some modification [16]. Briefly, sera samples were heat-inactivated for 30 min at 56 °C, then serially diluted two-fold, starting at 1:5 with DMEM containing 2% FBS in 96-well micro-plates, and then equal volume mixed with 100 TCID_50_ of SA 14-14-2 virus and incubated for 2 h at 37 °C. The virus-sera mixtures were transferred to 96-well plates containing confluent Vero cell monolayers at 37 °C for 1 h, and then removed the virus/sera-containing medium followed by washing with PBST, then DMEM medium containing 2% FBS and 0.5% agarose was added to each well and incubated for 3 days at 37 °C. The cells were fixed with 10% formalin and stained with 0.2% crystal violet. Plates were washed with ultra-pure water and air dried, followed by plaques counting. Neutralization titer was determined by calculating the reciprocal of the highest dilution ratio of sera that protected 50% of cells from infection through a variable-slope sigmoidal dose response computer model. 

### 2.8. ELISPOT

JEV-specific T cells response in mouse splenocytes was assessed by IFN-γ ELISPOT assays using pool of overlapping 20-mer peptides spanning the entire JEV Env protein as previously described [17]. Next, 96-well plates were coated with 100 μL per well of 10 μg/mL anti-mouse IFN-γ (BD Biosciences, San Jose, CA, USA) overnight at 4 °C, and then blocked for 1 h with sterile PBST containing 5% FBS at 37 °C, washed three times with sterile PBST. Splenocytes were resuspended to 2 × 10^5^ cells /mL in RPMI 1640 medium with 10% heat-inactivated FBS (Invitrogen), and then added to the plate and subsequently stimulated with Env peptide pool (2 μg/mL individual), concanavalin A (10 mM) as the positive control, and RPMI 1640 medium as the negative control, respectively. Following 20 h incubation at 37 °C, the plates were washed five times with sterile PBST and the last time with distilled water. The plates were then incubated with 2 μg/mL biotinylated antimouse IFN-γ for 1 h at room temperature, washed six times with sterile PBST and incubated for 1 h at room temperature with streptavidin–HRP, following five washes with sterile PBST and one with sterility PBS, the plates were added AEC substrate solution (Abcam, Cambridge, UK), when the spots were clear enough, the reaction was stopped by washing with distilled water, air dried, and spots were counted with an ELISPOT Analysis System (At-Spot-2100, AntaiYongxin, Beijing, China). Cells alone in the absence of stimulant were used as a negative control. The numbers of spot-forming cells (SFC) per 10^6^ cells were calculated.

### 2.9. Adoptive Transfer Experiment

The sera of BALB/c mice were collected and pooled one week after the final vaccination. Two groups of 4–5-week-old male and female A129 IFN-α/β receptor-deficient (*ifnar1^−/−^*) mice (*n* = 5 per group) were infused via tail intravenous injection (IV) with 0.2 mL sera from JEV VLP, or PBS vaccinated mice, after 1 h, mice were challenged by the intraperitoneal (i.p.) route with 1 × 10^5^ TCID_50_ JEV SA 14-14-2 strain. Sera were collected daily before and after challenge within 6 days for viremia determination using reverse transcription and TaqMan quantitative PCR (qRT-PCR). Clinical symptoms, bodyweight change rate and survival rate were monitored within 15 days.

### 2.10. qRT-PCR

Viral loads in mice sera were determined by qRT-qPCR. Viral RNA was extracted from sera using AxyPrep Total RNA Miniprep Kit (CORNING, USA). qRT-qPCR was performed using the One Step PrimeScript RT-PCR Kit (Takara, Japan) on an ABI 7500 Real-Time PCR System. Primers and probe were specific for the gene encoding prM/Env, as follows: F: 5’- TGGCAGTAACAACGGTCAA-3’, R: 5’- GTCTCCTTCTAGCACCAAGTC-3’, probe: 5’-FAM-CATCCTCCTGCTGTTGGTCGCTCCG-BHQ1-3’. The 20 μL reaction mixtures were set up with 8μL of total RNA. Cycling conditions were as follows: 42 °C for 5 min, 95 °C for 10 s, followed by 40 cycles of 95 °C for 5 s and 55 °C for 30 s. The JEV prM/Env gene was cloned into pET15a vector (Novagen, Billerica, MA, USA) and in vitro transcribed to RNA using T7 RNA polymerase (Promega, Madison, WI, USA). RNA transcripts were further digested with RNase-free DNase I (Promega, Madison, WI, USA) to remove residual plasmid DNA. After purification using the RNeasy cleanup kit (Qiagen, Hilden, Germany), RNA concentration was determined by a Nano-Drop 2000 spectrophotometer (Thermo Fisher Scientific, Waltham, MA, USA). Serial dilutions of RNA standard were included with each qRT-PCR to create a standard curve. Viral loads were expressed on Log dilutions of the RNA as viral copies/mL after calculation with the standard curve. The limit of detection was 1000 copies/mL.

### 2.11. Statistics

Statistical significance among different groups was determined as described in the text. The one-way or two-way analysis of variance followed by Tukey–Kramer multiple comparison tests, two-sided log-rank tests, Tukey–Kramer multiple comparison tests, and Student’s *t*-test were used in some experiments with probability (*p*) value < 0.05 considered to be statistically significant.

## 3. Results

### 3.1. Construction and Characteristics of JEV VLP

The JEV prM/Env lacking transmembrane region was designed and constructed (Figure 1a). The intracellular expression plasmid bearing prM/Env gene was transformed into *Pichia*
*pastoris* and identified by PCR. Then, SDS-PAGE and Western blot assays for the methanol-induced positive samples showed that prM/Env was successfully expressed with the molecular size of about 52 kDa (Figure 1b,c). The molecular weight of overexpressed protein is slightly larger than expected for it was glycosylated. After purification, the protein was verified by N-terminal sequencing, which showed that the N-terminal 33 aa prM-peptide was cleaved off efficiently from native Env protein. Then the protein was assembled into VLP in vitro and further verified using negative straining electron microscopy. The VLP formed by the expressed proteins were 20–30 nanometers (nm) in diameter. The primary particles were approximate 30 nm (Figure 1d).

### 3.2. JEV VLP Elicited Robust Humoral and T Cell Immune Responses in BALB/c Mice 

To assess the immunogenicity of JEV VLP, 6-week-old BALB/c female mice were divided into three groups, which were immunized three times with JEV VLP, SA 14-14-2 live-attenuated vaccine, or PBS control s.c. at a 21-day interval (Figure 2a). Serum samples were collected for binding antibodies (IgG) (ELISA) and NAbs (PRNT_50_) test. The results showed that JEV-specific ELISA titers in the immunization groups gradually increased and peaked after the final immunizations (Figure 2b). In addition, the IgG antibodies could be detected in the VLP group after the first dose, while they could be detected in the live-attenuated vaccine group after boost immunization. For the NAbs, the geometric mean titers (GMTs) of VLP-inoculated group reached 205.4 (95% CI: 145.9–289.1) at three weeks after the second dose and increased to 1384 (95% CI: 871–2199) after the final immunization (Figure 2c). Moreover, there was no significant difference in NAbs level between VLP group and live-attenuated vaccine group. In contrast, the ELISA and NAb titers were not detected in the negative control group. 

To further determine the cellular immune response elicited by JEV VLP, spleen lymphocytes from the mice were harvested at seven days after the third dose, and T cell responses were measured by IFN-γ ELISPOT. The significantly higher levels of INF-γ secreted by splenocytes were detected by stimulation with Env peptide pools (463 ± 42 spots/10^6^ cells, Figure 2d) compared to the control group. In addition, although SA 14-14-2 induced a more robust T cell response than VLP, there was no significant difference between the two groups.

### 3.3. Protection of Immunocompromised Mice through Passive Transfer of Anti-VLP Sera

To assess the protective efficacy of the antibodies in BALB/c mice induced by JEV VLP, the antisera collected at seven weeks after the last immunization were adoptively transferred to 4–5-week-old A129 IFN-α/β receptor-deficient mice (200 μL per mouse) via IV route. One hour later, each A129 mouse was IP challenged with a lethal dose of JEV SA 14-14-2 strain (Figure 3a). The results showed that all the mice receiving antisera were normal with the gradually increasing body weights and without any clinical syndromes, as well as viral RNA loads in the sera within six or eight days postinfection (DPI) (Figure 3b,c). However, all the control mice died during 7–9 DPI (Figure 3d), and viral RNA loads were detected as early as one DPI till the last detection time point at six DPI, indicating viremia existed at least within six DPI. These data indicated that antisera conferred complete passive protection to interferon receptor-deficient mice against JEV challenge.

### 3.4. JEV VLP Elicited Robust Humoral Immune Response in Pigs

To determine the immunogenicity of VLP in JEV host animal, as well as the minimal inoculation dosage, a series of dosages with 2.5, 5, 10, and 20 μg were used to immunize IM 4-week-old piglets at a 21-day interval, respectively (Figure 4a). The JEV-specific ELISA (Figure 4b) and NAbs (Figure 4c) were tested, and both titers in all groups were gradually increased after the first or second immunization and reached the plateaus after three weeks of the final immunization. The pigs immunized with 5, 10, and 20 μg VLP displayed significant higher titers than that elicited by 2.5 μg VLP during the entire experiment (Figure 4b,c). The NAb GMTs in the plateaus were similar in the 5, 10, and 20 μg groups with 218 (95% CI: 124.7–369.4), 328.2 (95% CI: 211.1–499.7), and 481.3 (95% CI: 318.6–714.7), while 2.5 μg group was only reached 12.09 (95% CI: 5.97–23.23). Serum neutralization titer of 10 is usually considered sufficient for protection against JEV [18], suggesting that pigs receiving doses above 5 μg are able to develop protective immunity. However, the neutralization titer of 5 μg group with 18.83 (95% CI: 13.36–26.54) was just above the minimum protective antibody level after the second dose. Therefore, it was necessary to immunize three times at a dose of 5 μg to obtain lasting protection. In contrast, no ELISA or NAb titers were detected in control pigs. In conclusion, JEV VLP can induce a protective humoral immune response in pigs, suggesting it could be as the JEV vaccine candidate.

## 4. Discussion

JE continues to be a crucial infectious disease in Asia, and vaccination was proven to be effective in reducing the incidence. Although several authorized vaccines are available, there is still a need for safe, effective, and affordable vaccines in JEV endemic areas. Multiple strategies were used to develop JEV vaccines, including inactivated vaccines [19,20], live-attenuated vaccines [14,16,21], recombinant vector vaccines [22], nucleic acid vaccines [8,9,10], and subunit vaccines [11,13,23]. Vaccines developed by different approaches have distinct advantages and limitations. As the second-generation vaccines, subunit vaccines are selected by more and more research teams, and the VLP subunit vaccines attracted more attention because of their unique advantages [11,13].

VLPs are multimeric or multiprotein nanostructures that are assembled to form virion structures but lack genetic material [24]. The conformational epitope of VLPs is almost identical to the natural virus. So VLPs could display high-density B-cell and T-cell epitopes and induce strong humoral and cellular immunity without adjuvant. Some VLP vaccine candidates against hepatitis B virus (HBV), hepatitis D virus (HDV), human papillomavirus (HPV), West Nile virus (WNV), Chikungunya virus (CHIKV), and Dengue virus (DENV) were reported [25,26,27,28,29], as well as JEV [11,13]. The VLP vaccines which showed good safety and strong immunogenicity, have been produced using different expression systems, including bacterial systems, yeast systems, insect systems, plant systems, mammalian cell systems, cell-free systems [30]. The bacterial systems have the advantages of fast reproduction and low production cost. However, due to the lack of necessary protein modification, it is rarely used at present. Insect and mammalian expression systems are capable of post-translational modification of exogenous proteins. The previously reported JEV VLP vaccine candidates were produced by expression in insect or mammalian cells [11,13]. However, researchers are still pursuing cheaper and easier-to-scale preparation systems due to cost and yield constraints.

The yeast system is a successful foreign protein expression system with the advantages of both prokaryotic and eukaryotic expression systems. It has a higher yield, lower production cost, and necessary protein post-translational modification characteristics. It was used to produce several VLP vaccines, including HBV, HDV, EV71, and HPV [31,32,33]. At present, the yeast used in vaccine production mainly includes *Saccharomyces cerevisiae*, *Hansenula* and *Pichia pastoris*. As a food-grade yeast, *Saccharomyces cerevisiae* has good safety. The licensed HBV (Recombivax HB, Merck, Darmstadt, Germany) and HPV (Gardasil^®^, Merck, Darmstadt, Germany) vaccines derived from *Saccharomyces cerevisiae* were widely vaccinated. *Hansenula* and *Pichia pastoris* show faster reproduction speed or stronger expression regulation characteristics. Therefore, many vaccines in clinical trials are produced by *Hansenula* or *Pichia pastoris* [34,35]. However, the current VLP vaccines expressed by yeast are mainly nonenvelope viruses, and whether JEV, as an enveloped virus, can generate VLPs is still unknown. We were surprised to find that the recombinant JEV Env protein can be independently assembled into spherical VLP in vitro. Its structure is similar to that of the natural virus. This discovery also provides a new idea for developing other enveloped virus VLP vaccines.

The JEV VLP based on the *Pichia pastoris* expression system showed good immunogenicity and protective efficacy and had good application prospects. Moreover, the T cell immune response is critical in virus clearance and clinical symptom alleviation [36,37]. JEV VLP can not only induce high levels of neutralizing antibodies, but also induce a robust cellular immune response, which is lacking in conventional subunit vaccines; therefore, JEV VLP has an immune effect similar to that of live-attenuated vaccine. Besides, there is no potential biosafety risk. A JE neutralizing antibody titer >1:10 is commonly accepted as evidence of protection [18]. The VLP vaccine could make JEV host animal domestic pigs obtain neutralizing antibody immune protection (NAb titers > 10) after two doses, and a second booster immunization can further increase the antibody level significantly. Furthermore, we found that the immune response induced by VLP alone was more potent than that of the formula with aluminum adjuvant (data not shown). In addition, we found that JEV VLP is very sensitive to the pH and ionic strength of the buffer. Dissociation of VLP occurs when these parameters change. Therefore, we believe that the addition of adjuvant resulted in the change of pH and ionic strength of VLP solution, thus destroying the structure of VLP and reducing its immunogenicity. In addition, unlike HPV, JEV belongs to envelope virus. Due to the lack of lipid membrane support, the stability of JEV VLP may be relatively poor. Therefore, to maintain the structural integrity of JEV VLP, we should not use adjuvant or choose an adjuvant that does not destroy VLP.

To evaluate the protective efficacy of VLP candidate vaccine more conveniently, we refered to the previously reported immune-deficient mouse infection model based on attenuated vaccine strain SA14-14-2, which requires only BSL-2/ABSL-2 containment [38]. This model selected AG129 mice deficient in interferon α/β and γ receptors, but we found that AG129 is easy to die accidentally and is not easy to obtain. Therefore, we replaced it with A129 mice. Unlike AG129 mice, A129 mice were not knocked out of the interferon γ receptor, so their innate immune system is stronger. In addition, we also increased the virus challenge dose from 10 Pfu to 10^5^ TCID_50_. The results proved that this modified JEV infection model is feasible.

## 5. Conclusions

Our VLP subunit vaccine candidate induced robust humoral and cellular immune responses and conferred complete protection against JEV infection in mice. There were no abnormalities in bodyweight, viremia, or clinical signs after the challenge. These data proved the possibility that the VLP is a promising candidate vaccine to prevent JEV infection.

## Figures and Tables

**Figure 1 vaccines-10-00197-f001:**
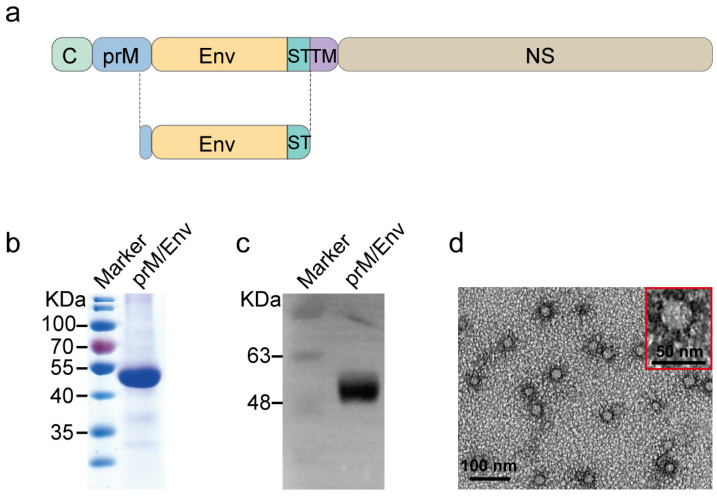
Construction and characterization of Japanese encephalitis virus (JEV) virus-like particle (VLP). (**a**) Schematic design of JEV VLP vaccine candidate. (**b**) purified recombinant prM/Env protein was analyzed by SDS-PAGE. (**c**) Immunoblot analysis of recombinant prM/Env protein with polyclonal antibodies against JEV. (**d**) VLP shape and size were tested by electron microscopy with negative staining method. spherical VLP size was 20–30 nm in diameter and mainly in ~30 nm. Scale bar: 100 nm, 50 nm.

**Figure 2 vaccines-10-00197-f002:**
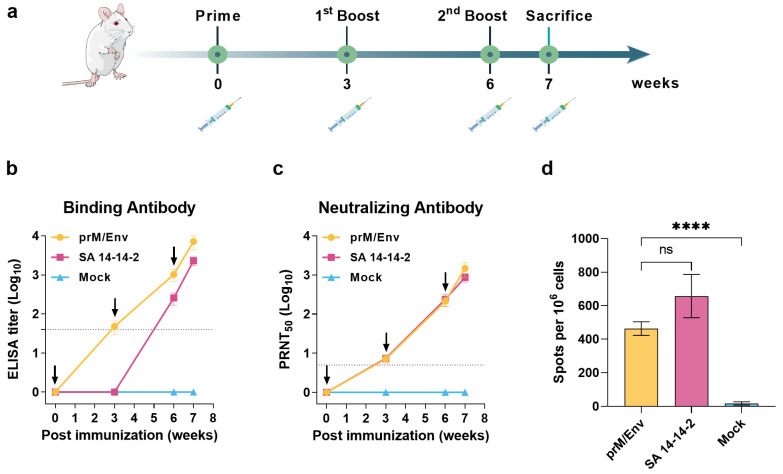
JEV VLP induced robust humoral and T cell immune responses in BALB/c mice. (**a**) Flow chart of JEV VLP immunogenicity evaluation in mice model. Solid vertical lines indicate weeks of immunization (black), sacrifice (green), and syringe symbols indicate blood sampling time points. (**b**) JEV specific binding antibodies were assessed by ELISA. Dotted lines indicate detection limit. (**c**) Neutralizing antibodies against JEV SA 14-14-2 strain were measured by PRNT assay. Dotted lines indicate detection limit. (**d**) Spleens and splenocytes were collected at weeks 7 for IFN-γ detection by ELISPOT after stimulation with JEV Env peptide pools. Data are shown as means ± SEM (standard errors of means). *p* values were analyzed by Student’s *t*-test (ns, *p* > 0.05; ****, *p* < 0.0001).

**Figure 3 vaccines-10-00197-f003:**
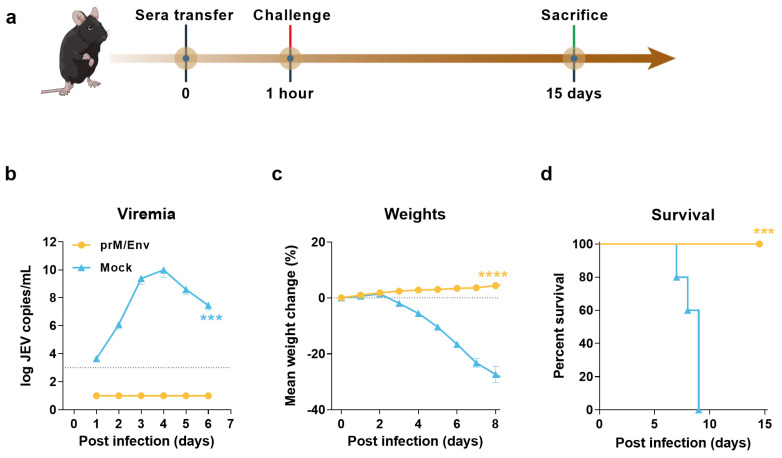
Passive protection of anti-VLP sera in immunocompromised mice model. (**a**) Flow chart of experimental design. Each A129 mouse was given 200 μL sera from JEV VLP or PBS vaccinated mice via IV route. After 1 h of injection, each mouse was infected via i.p. with 1 × 10^5^ TCID_50_ of SA 14-14-2 strain. Solid vertical lines indicate timeline of immunization (black), challenge (red), and sacrifice (green). Serum collected within 6 DPI were used for viral loads detection by qRT-PCR (**b**). Dotted lines indicate detection limit. Bodyweight changes (**c**) and survival rates (**d**) of mice in all groups were monitored daily after challenge. Data are shown as means ± SEM. A two-way analysis of variance (ANOVA) with Tukey’s multiple-comparison tests was performed to statistically analyze significant difference of viral loads between JEV VLP and PBS-Mock. Weight change was analyzed by Tukey-Kramer multiple comparison test. Survival data were analyzed by two-sided log-rank test. (***, *p* < 0.001; ****, *p* < 0.0001).

**Figure 4 vaccines-10-00197-f004:**
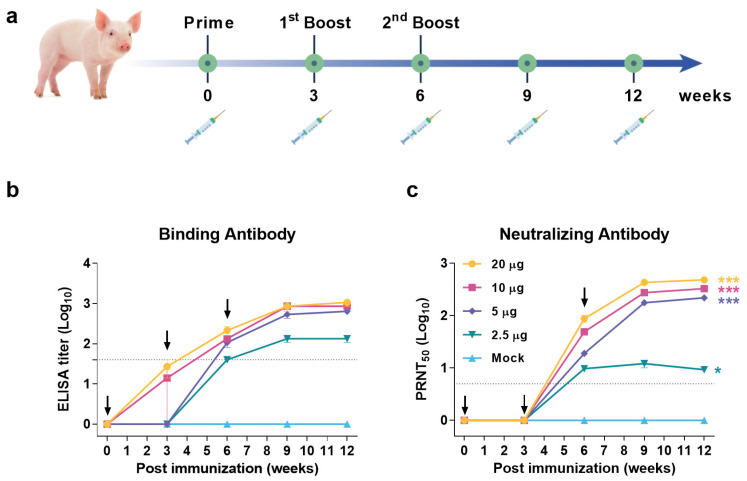
Humoral immune response elicited by different doses of JEV VLP in pigs. (**a**) Flow chart of JEV VLP immunogenicity evaluation in pigs. Groups of 4-week-old pigs were immunized three times by IM route with 2.5 μg, 5 μg, 10 μg, or 20 μg JEV VLP, or PBS as control. Solid vertical lines indicate weeks of immunization (black), and syringe symbols indicate blood sampling time points. (**b**) JEV specific binding antibodies were assessed by ELISA. (**c**) Neutralizing antibodies against JEV SA 14-14-2 strain were measured by PRNT assay. Dotted lines indicate detection limit. Data are shown as means ± SEM. NAb differences between immunized and control groups were analyzed by Tukey–Kramer multiple comparison test. (*, *p* < 0.05; ***, *p* < 0.001).

## Data Availability

The data that support the findings of this study are available from the corresponding author, L.Y., upon reasonable request.

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
