# Peer review of "A VLP-Based Vaccine Candidate Protects Mice against Japanese Encephalitis Virus Infection"

_vaccines, 2022, doi:10.3390/vaccines10020197_

Round 1

Reviewer 1 Report

This is a very interesting study that has the potential to develop a new vaccine for Japanese encephalitis virus (JEV). They provided convincing data to prove the strong immunogenicity of their hybrid JEV prM/Env VLPs expressed in Pichia pastoris. They monitored neutralizing antibodies in vaccinated mice and pigs and also tested protection against the lethal challenge of JEV in A129 mice using a passive immunization strategy. They further demonstrated a dose related (>5ug) immune response in pigs. 
I wish they could provide more details in the introduction about what specific "side effects" of the current vaccines that prevent them from being accepted by people all over the world. 
VLPs are very immunologic by itself. In current VLPs based vaccines, different adjuvants have been used to argument immunogenicity of VLPs. Line 371-375,  the authors observed that aluminum adjuvants reduced the immunogenicity of their VLPs. They believe that the adjuvant damages the structure of VLPs. I have several questions: 
1) Which aluminum adjuvant was used and how was it formulated for this test?
2) Did they test this formulated VLPs by TEM or ELISA using antibodies against conformational epitopes?
3) Do they have any explanation why aluminum adjuvant did not interfere with other VLPs such as HBV or HPV?  or prM/Env VLPs are very unstable? 

Author Response

  1. I wish they could provide more details in the introduction about what specific "side effects" of the current vaccines that prevent them from being accepted by people all over the world. 

Response: Thanks for your comments and constructive suggestions. Event Reporting System (VAERS) received 300 adverse event reports following inactivated mouse brain-derived JE vaccination (24 per 100,000 doses distributed) from 1999 to 2009, and hypersensitivity reactions were common among persons receiving inactivated mouse brain-derived JE vaccine. Live attenuated vaccines have been widely used in Asian countries, and no obvious side effects have been reported so far. Developed countries such as the United States and the European Union have not approved live attenuated vaccines, mainly because they believe that live attenuated vaccines, as a virus that can proliferate, have potential biosafety risks. Moreover, inactivated vaccines do not have this risk compared to live attenuated vaccines. Therefore, these countries have mainly adopted inactivated vaccine strategies. According to your suggestion, we have added this explanation in the Introduction section. Please see lines 49–52.

  1. VLPs are very immunologic by itself. In current VLPs based vaccines, different adjuvants have been used to argument immunogenicity of VLPs. Line 371-375, the authors observed that aluminum adjuvants reduced the immunogenicity of their VLPs. They believe that the adjuvant damages the structure of VLPs. I have several questions: 

1)  Which aluminum adjuvant was used and how was it formulated for this test?

Response: We chose a commercial aluminum hydroxide adjuvant “Imject® Alum” (Thermo, Cat. #77161). The JEV-VLP was mixed with an equal amount of aluminum hydroxide adjuvant.

2)  Did they test this formulated VLPs by TEM or ELISA using antibodies against conformational epitopes?

Response: We initially evaluated the immunogenicity of the aluminum-adjuvanted JEV-VLP vaccine candidate in a mouse model, but the results were unsatisfactory. So we tried to immunize JEV-VLP alone, and were surprised to find that the antibody level was significantly increased. We thought that aluminum adjuvant should destroy the structure of VLP, and the results of TEM analysis confirmed our inference. The results showed that the number of intact VLPs was significantly reduced after adding aluminum adjuvant. Since we did not have a JEV conformational epitope mAb, JEV-VLP conformational epitope analysis was not performed by the ELISA method.

3)  Do they have any explanation why aluminum adjuvant did not interfere with other VLPs such as HBV or HPV?  or prM/Env VLPs are very unstable? 

Response: We found that JEV-VLP is very sensitive to the pH and ionic strength of the buffer. Dissociation of VLP occurs when pH and ionic strength change. Therefore, we believe that the addition of adjuvant resulted in the change of pH and ionic strength of VLP solution, thus destroying the structure of VLP. In addition, unlike HPV, JEV belongs to envelope virus. Due to the lack of lipid membrane support, the stability of JEV-VLP may be relatively poor. However, since this study was only an exploratory work, we demonstrated the feasibility and efficacy of the JEV-VLP vaccine candidate, and also found that its stability was not good enough. We will conduct in-depth research on its stability in the future, hoping to find an adjuvant formulation that can maintain its structural stability. According to your comments, we have added this explanation in the Discussion section. Please see lines 391–397.

Reviewer 2 Report

Manuscript titled A VLP-based Vaccine Candidate Protects Mice against Japanese Encephalitis virus Infection by Yang et al is a nice piece of work. Overall draft is OK but need a minor revision before it can be accepted for publication.

My specific comments to author are given below

Thorough check of writing throughout the draft.

In page 3, line 126, is it 15V or 150V, please correct it

Some sentences are from abstract is repeated in other section of draft. please take care of it.

In discussion section line 330, please modify sentence as it contradicts, as measures to control or prevent disease is already present.

I think author should include 1-2 paragraph either in introduction or in discussion about yeast-based vaccines (which is a key aspect of this paper) and should include some important papers.

Author said that size of VLPs are around 30 nm (form EM data), but should also perform DLS to confirm the size of VLPs

Also need to mention procedure for EM for size estimation or should include suitable reference if procedure is adopted from somewhere.

EM image in Fig 1 should be zoom for clarity

Author Response

  1. Thorough check of writing throughout the draft. In page 3, line 126, is it 15V or 150V, please correct it. Some sentences are from abstract is repeated in other section of draft. please take care of it. In discussion section line 330, please modify sentence as it contradicts, as measures to control or prevent disease is already present.

Response: Thanks for your comments. We have made some corresponding modifications according to your suggestion.

  1. I think author should include 1-2 paragraph either in introduction or in discussion about yeast-based vaccines (which is a key aspect of this paper) and should include some important papers.

Response: According to your suggestion, we have added a summary of yeast-based vaccines in the Discussion section. Please see lines 359–369.

  1. Author said that size of VLPs are around 30 nm (form EM data), but should also perform DLS to confirm the size of VLPs

Response: Since the size of the JEV VLPs we obtained was not uniform, but a mixture of spherical particles with different diameters. Although we also analyze the VLP by dynamic light scattering, considering that the electron microscope results are more intuitive, we still use the ruler of electron microscope photos to estimate the size of VLPs.

  1. Also need to mention procedure for EM for size estimation or should include suitable reference if procedure is adopted from somewhere.

Response: We estimated the VLP size through the scale on the electron microscope photo.

  1. EM image in Fig 1 should be zoom for clarity

Response: According to your suggestion, a zoomed electron microscope image is added to Fig. 1d.